# PGC-1α Regulates Cell Proliferation, Migration, and Invasion by Modulating Leucyl-tRNA Synthetase 1 Expression in Human Colorectal Cancer Cells

**DOI:** 10.3390/cancers15010159

**Published:** 2022-12-27

**Authors:** Jun Gi Cho, Su-Jeong Park, Sang-Heum Han, Joo-In Park

**Affiliations:** 1Department of Biochemistry, Dong-A University College of Medicine, Busan 49201, Republic of Korea; 2Department of Translational Biomedical Sciences, Graduate School, Dong-A University, Busan 49201, Republic of Korea; 3Peripheral Neuropathy Research Center, Dong-A University, Busan 49201, Republic of Korea

**Keywords:** PGC-1α, leucyl-tRNA synthetase 1 (LARS1), AKT, invasion, colorectal cancer

## Abstract

**Simple Summary:**

There are still controversies about the roles of peroxisome proliferator-activated receptor γ coactivator 1α (PGC-1α) and leucyl-tRNA synthetase 1 (LARS1) in cancer. In this study, we examined whether the effects of PGC-1α on cell proliferation and invasion were mediated by modulation of LARS1. Our results showed that PGC-1α regulated cell proliferation and invasion by regulating the LARS1/AKT/GSK3β/β-catenin axis in human colorectal cancer cells. These data suggest that LARS1 might be a potential therapeutic target for PGC-1α-overexpressing human colorectal cancer.

**Abstract:**

Although mounting evidence has demonstrated that peroxisome proliferator-activated receptor γ coactivator 1α (PGC-1α) can promote tumorigenesis, its role in cancer remains controversial. To find potential target molecules of PGC-1α, GeneFishing^TM^ DEG (differentially expressed genes) screening was performed using stable HEK293 cell lines expressing PGC-1α (PGC-1α-HEK293). As results, leucyl-tRNA synthetase 1 (LARS1) was upregulated. Western blot analysis showed that LARS1 was increased in PGC-1α overexpressed SW480 cells but decreased in PGC-1α shRNA knockdown SW620 cells. Several studies have suggested that LARS1 can be a potential target of anticancer agents. However, the molecular network of PGC-1α and LARS1 in human colorectal cancer cells remains unclear. LARS1 overexpression enhanced cell proliferation, migration, and invasion, whereas LARS1 knockdown reduced them. We also observed that expression levels of cyclin D1, c-Myc, and vimentin were regulated by LARS1 expression. We aimed to investigate whether effects of PGC-1α on cell proliferation and invasion were mediated by LARS1. Our results showed that PGC-1α might modulate cell proliferation and invasion by regulating LARS1 expression. These results suggest that LARS1 inhibitors might be used as anticancer agents in PGC-1α-overexpressing colorectal cancer. Further studies are needed in the future to clarify the detailed molecular mechanism by which PGC-1α regulates LARS1 expression.

## 1. Introduction

Peroxisome proliferator-activated receptor γ coactivator-1α (PGC-1α), one of the nuclear receptor coactivators, is known to regulate the metabolic pathway via interactions with several transcription factors [1]. It was first identified to interact with peroxisome proliferator-activated receptor γ (PPARγ) in brown adipocytes [2]. It also interacts with and enhances the activity of transcription factors such as nuclear respiratory factor-1/2 (NRF1/2), yin yang 1 (YY1), and estrogen-related receptors (ERRs), which are involved in the control of mitochondrial biology [3]. Accumulating evidence demonstrated that PGC-1α has dual functions, such as antitumor effects and tumor-promoting effects, depending on the cell context and tumor type [4,5,6,7,8,9,10]. PGC-1α null mouse with engineered tumor-prone backgrounds promoted colorectal and prostate carcinoma development [5,6]. In contrast, several papers showed PGC-1α had a tumor-promoting effect [7,8,9,10]. Our previous studies have shown that PGC-1α has tumor-promoting effects through multiple mechanisms, such as the upregulation of Specificity protein 1 (Sp1), acyl-CoA binding protein (ACBP), fatty acid synthase (FASN), and the activation of the AKT/Glycogen synthase kinase-3β (GSK-3β)/β-catenin pathway [8,9,10]. Even though intensive investigation was performed, the role and detailed underlying molecular mechanisms of PGC-1α in cancer remains controversial.

Aminoacyl-tRNA synthetases (aaRSs) catalyze the binding of amino acids to their respective tRNAs and play an important role in the maintenance of cell survival [11]. Among them, leucyl-tRNA synthetase 1 (LARS1) catalyzes the covalent binding of leucine to tRNA_Leu_ during polypeptide synthesis. It also acts as a leucine sensor in activating the mechanistic target of rapamycin complex 1 (mTORC1) [12,13]. mTORC1 is a serine/threonine kinase, integrating signals from amino acids, growth factors, and the energy level of the cells, and plays an important role in cell growth and protein synthesis [14,15]. It can phosphorylate its downstream targets, ribosomal S6 kinase 1 (S6K1) and eukaryotic initiation factor 4E binding protein 1 (4EBP1), regulating protein synthesis [16]. A previous study demonstrated that knockdown of LARS1 inhibits cell migration and colony-forming ability in lung cancer [17]. In addition, it was reported that inhibition of LARS1 may reduce cancer cell proliferation via the p21 signaling pathway and cause apoptosis [18]. In recent, many investigators have focused on LARS1 as a potential target of anticancer agents [18,19,20,21,22,23,24]. However, its role and the molecular mechanism in colorectal cancer are not clearly defined yet.

Until now, the molecular network of PGC-1α and LARS1 in human colorectal cancer cells are not explored. Thus, we aimed to investigate whether PGC-1α actions on cell proliferation and invasion are mediated by LARS1.

## 2. Materials and Methods

### 2.1. Cell Cultures

Human embryonic kidney 293 (HEK293), SW480, SW620, HT-29, and SNU-C4 (human colorectal cancer) cells (Korean Cell Line Bank, Seoul National University, Seoul, Republic of Korea) were grown in Dulbecco’s modified Eagle’s medium (DMEM) (Hyclone, Logan, UT, USA) supplemented with 10% fetal bovine serum (FBS) (Gibco, Carsbad, CA, USA) and penicillin/streptomycin (Hyclone, Logan, UT, USA). They were maintained in a humidified atmosphere of 95% air/5% CO_2_ at 37 °C. pcDNA-HEK293 cells and PGC-1α overexpressed HEK293 (PGC-1α-HEK293) cells were previously established [8] and used in this experiment.

### 2.2. Materials

Crystal violet was obtained from Sigma-Aldrich (St. Louis, MO, USA). Antibodies against AKT (#4685), phospho-AKT (Ser473) (p-AKT; #4060), phospho-mTOR (Ser2448) (p-mTOR; #5536), mTOR (#2983), phospho-S6 kinase (Thr389)(p-S6K; #9205), S6K (#2708), phospho-4EBP1 (The37/46) (p-4EBP1; #2855), 4EBP1 (#9644), cyclin D1 (#2926), GSK-3β (#9332), and phospho-GSK-3β (Ser9)(p-GSK-3β; #9336) were purchased from Cell Signaling Technology (Danvers, MA, USA). Anti-vimentin antibody (ab92547) and anti-c-Myc antibody were obtained from Abcam (Cambridge, UK). Antibodies against phospho-β-catenin (p-β-catenin; sc-16743R) and β-catenin (sc-7199) were bought from Santa Cruz Biotechnology (Santa Cruz, CA, USA). Antibody against PGC-1α (A12348) was purchased from ABclonal Biotech Co., Ltd. (Boston, MA, USA). Antibody against LARS1 (21146-1AP) was purchased from Proteintech Group, Inc. (Rosemont, IL, USA). Anti-β-actin (A1978), anti-rabbit IgG (A0545), and anti-mouse IgG secondary antibodies (A9044) were purchased from Sigma-Aldrich (St. Louis, MO, USA). Unless otherwise stated, all other chemicals were purchased from Sigma-Aldrich (St. Louis, MO, USA).

### 2.3. RNA Isolation and First-Strand cDNA Synthesis

Total RNA isolation and first-strand cDNA synthesis were performed as described previously [8]. In brief, total RNAs were isolated from pcDNA-HEK293 cells and PGC-1α-HEK293 cells using an RNAeasy mini kit (Qiagen, Valencia, CA, USA). DNase I treatment of total RNAs and reverse transcription were then performed to obtain first-strand cDNAs using GeneFishing^TM^ DEG kits (Seegene, Seoul, Republic of Korea) following the manufacturer’s protocol. First-strand cDNAs were diluted by adding 80 μL of ultra-purified water for GeneFishing^TM^ PCR (Seegene, Seoul, Republic of Korea). They were stored at −20 °C until use.

### 2.4. Annealing Control Primer (ACP)-Based GeneFishing PCR

Differentially expressed genes between pcDNA-HEK293 cells and PGC-1α-HEK293 cells were screened using the ACP-based PCR method [25] with GeneFishing DEG kits (Seegene, Seoul, Republic of Korea). In brief, second-strand cDNA synthesis was performed at 50 °C during one cycle of first-stage PCR in a final reaction volume of 20 μL containing 3–5 μL (~50 ng) of diluted first-strand cDNA, 1 μL of dT-ACP2 (10 μM), 1 μL of 10 μM arbitrary ACP, and 10 μL of 2X Master Mix (Seegene, Seoul, Republic of Korea). In total, 120 different types of arbitrary ACPs were used. The PCR condition for second-strand synthesis was one cycle at 94 °C for 1 min, followed by 50 °C for 3 min and 72 °C for 1 min. After second-strand DNA synthesis was completed, the second-stage PCR amplification was performed using 40 cycles of 94 °C for 40 s, followed by 65 °C for 40 s and 72 °C for 40 s and a 5-min final extension at 72 °C. Amplified PCR products were separated on a 2% agarose gel stained with ethidium bromide. Amplified cDNA fragments with >2-fold differential band intensities were re-amplified and extracted from the gel using a GeneClean II kit (Qbiogene, Solon, OH, USA) and directly sequenced with an ABI PRISM 310 Gene Analyzer (Applied Biosystems, Waltham, MA, USA).

### 2.5. Generation of Stable PGC-1α-Overexpressing SW480 and PGC-1α shRNA-Knocked Down SW620 Cell Line

To establish a stable PGC-1α-overexpressing SW480 cell line, cells were transfected with 4 μg of empty vector (pcDNA3.1) or pcDNA3.1-FLAG-PGC-1α expression vector from Spiegelman using Lipofectamine 2000 (Invitrogen, Carlsbad, CA, USA) following the manufacturer’s recommended procedure. After transfection, stable cell lines (PGC-1α-1- and PGC-1α-2-SW480 cells) were established after G418 selection (800 μg/mL) for 14 days. To obtain a stable PGC-1α-knockdown cell line, SW620 cells were transfected with 4 μg of nontargeting control (NC) shRNA or PGC-1α shRNA plasmid (KH00461N; Qiagen, Hilden, Germany) using Lipofectamine 2000 (Invitrogen, Carlsbad, CA, USA) following the manufacturer’s instructions. After transfection, cells were treated with G418 (800 μg/mL) for 14 days and five clones were isolated. Among them, PGC-1α shRNA-1- and PGC-1α shRNA-2- SW620 cells were efficiently knocked-down and used in this study.

### 2.6. Generation of Stable LARS1-Overexpressing SW480 and LARS1 shRNA-Knocked Down SW620 Cell Line

To establish a stable LARS1-overexpressing SW480 cell line, cells were transfected with 4 μg of empty vector (pCMV6-AC-GFP; PS100010) or LARS1 expression vector (LARS1 human tagged ORF clone; RG221682) from Origene (Rockville, MD, USA) using Lipofectamine 2000 (Invitrogen, Carlsbad, CA, USA) following the manufacturer’s recommended procedures. After transfection, stable cell lines (LARS1-2- and -4-SW480 cells) were established after G418 selection (800 μg/mL) for 14 days. To obtain a stable LARS1-knockdown cell line, SW620 cells were transfected with 4 μg of nontargeting control (NC) shRNA (pGFP-V-RS shRNA vector; TR30007) or LARS1 shRNA plasmid (TG303577; Origene, Rockville, MD, USA) using Lipofectamine 2000 (Invitrogen, Carlsbad, CA, USA) following the manufacturer’s instructions. After transfection, cells were treated with G418 (800 μg/mL) for 14 days and six clones were isolated. Among them, LARS1 shRNA-4- and -5-SW620 cells were efficiently knocked-down and used in this study.

### 2.7. RNA Extraction and Real-Time Quantitative Reverse Transcriptase–Polymerase Chain Reaction (qRT-PCR)

Total RNA was extracted from cells using Trizol reagents (Invitrogen, Carlsbad, CA, USA) according to the manufacturer’s protocols. cDNA was synthesized with 50 ng RNA using M-MuLV transcriptase (M0253S; New England BioLabs, Ipswich, MA, USA) according to the manufacturer’s protocols. qRT-PCR was performed using a Power SYBR^®^ Green PCR Master Mix (4367659; Applied Biosystems, Waltham, MA, USA) on QuantStudio^TM^ 3 Real-Time PCR systems (Applied Biosystems, Waltham, MA, USA). PCR conditions are as follows: 95 °C for 10 min, 40 cycles of 95 °C for 15 s, 60 °C for 1 min. The following primers were used for PCR reactions: PGC-1α (forward primer: 5′- GACACAACACGGACAGAA-3′, reverse primer: 5′-CACAGGTATAACGGTAGGTAA-3′), LARS1 (forward primer: 5′-GGACAGCCTTGCATGGATCAT-3′, reverse primer: 5′- TAGATGGGTATGGCTCAAGCA-3′), β-Actin (forward primer: 5′-CGATGCCCTGAGGCTCTTT-3′, reverse primer: 5′-AGTGATGCCACAGGATTCCA-3′). Each mRNA expression levels were normalized with β-Actin and the average relative changes were calculated using the $2^{-△△Ct}$ method [26]. All reactions were carried out in triplicate.

### 2.8. Cell Counting

Cells were cultured at a density of 1.5 × 10^5^/well in 6-well plates. pCMV6-, LARS1-2, LARS1-4-SW480 cells, NC shRNA-, LARS1 shRNA-4, and LARS1 shRNA-5 SW620 cells were cultured for 24, 48, and 72 h, respectively. In addition, PGC-1α shRNA-SW620 cells after transfection with LARS1 shRNA or with NC shRNA were seeded at a density of 1.5 × 10^5^/well in 6-well plates and were incubated for 24, 48, or 72 h. Then, the cells were harvested by trypsinization using trypsin/EDTA and stained with trypan blue. A hemocytometer was used to count and calculate the average number of cells in each group. All experiments were repeated three times.

### 2.9. MTT Assay

pCMV6-, LARS1-2-, LARS1-4-SW480 cells, NC shRNA-, LARS1 shRNA-4-, and LARS1 shRNA-5-SW620 cells were seeded on 96-well plates (Corning Inc., Corning, NY, USA) at a density of 1 × 10^4^ cells/well in a volume of 200 µL and cultured for 48 h. MTT solution (5 mg/mL in PBS) was added to well (20 µL/well). Plates were incubated for an additional 4 h at 37 °C. MTT solution in the medium was removed by aspiration. To dissolve the formazan crystal formed in viable cells, 100 µL dimethyl sulfoxide (DMSO) was added to each well. Absorbance at 570 nm was then measured.

### 2.10. Transwell Migration and Invasion Assays

Transwell migration and invasion assays were performed as previously described [10]. For transwell migration assays, cells (1 × 10^5^) in serum-free medium were plated into the upper chamber of the inserts (24-well cell culture plate transwell inserts; 353097, Corning Inc., Corning, NY, USA) with 8 μm pore filters and complete medium was added to the lower chamber. After incubation for 48 h, the cells on the upper surface were removed with cotton tips and the cells migrated to the underside of the membrane were fixed with 4% formaldehyde in PBS for 30 min at RT, stained with 0.1% crystal violet (Sigma Aldrich, St. Louis, MO, USA) for 20 min and washed with PBS three times. For transwell invasion assays, cells (1 × 10^5^) were suspended in 200 μL of serum free medium and plated into the upper chambers of the inserts (24-well cell culture plate transwell inserts; 353097, Corning Inc., Corning, NY, USA) that were pre-coated with 50 μL of Matrigel (1 μg/μL; BD Biosciences, San Jose, CA, USA). The lower chamber had 600 μL of DMEM with 10% FBS added, and the plate was incubated for 48 h at 37 °C. After incubation, the cells on the upper surface were removed with cotton tips and the cells migrated to the underside of the membrane were fixed with 4% formaldehyde in PBS for 30 min at RT, stained with 0.1% crystal violet (Sigma Aldrich, St. Louis, MO, USA) for 20 min and washed with PBS three times. The number of cells was calculated from five random fields at a magnification of ×200, using an inverted microscope (Nikon Eclipse TS100; Nikon, Tokyo, Japan) and was indicated as the average number of cells/field of view. The mean value was calculated form three independent experiments.

### 2.11. Western Blot Analysis

Cell lysis and Western blot analyses were performed as described previously [10]. In brief, cells were lysed and electrophoresed on 8, 10, 12, or 15% sodium dodecyl sulfate-polyacrylamide gel electrophoresis (SDS-PAGE) and then blotted to polyvinylidene difluoride (PVDF) membranes (Merck Millipore, Darmstadt, Germany). Blots were blocked for 1 h in 5% skim milk in PBS at room temperature (RT). They were then probed with the appropriate primary antibody overnight at 4 °C. After washing, blots were probed with horseradish peroxidase-conjugated secondary antibodies for 2 h. After another wash, signals were detected with ECL (Amersham, Buckinghamshire, UK) according to the manufacturer’s instructions. Blots were also probed with a monoclonal anti-β-actin antibody, which served as an internal loading control. Bands were quantified using Image Studio Lite Ver 3.1. (LI-COR, Inc., Lincoln, NE, USA).

### 2.12. Immunofluorescence Staining and Fluorescence Quantification

Immunofluorescence staining was performed as previously described [10]. In brief, cells were cultured on Lab-Tek^®^ Chamber Slides^TM^ (Nalge Nunc, Inc., Rochester, NY, USA), and then fixed with formaldehyde (3%), permeabilized with Triton X-100 (0.3%), and blocked for 30 min with bovine serum albumin (BSA, 5%) at RT. Next, cells were washed with PBS and incubated with a series of primary antibodies as indicated, followed by staining with Alexa Fluor 488 conjugated donkey anti-rabbit IgG (#A-21206, Invitrogen, Carlsbad, CA, USA). Samples were then mounted using glycerol and analyzed using a laser confocal microscope (LSM800; Carl Zeiss, Jena, Germany) at the Neuroscience Translational Research Solution Center (Busan, Republic of Korea). Negative control staining was performed using only secondary antibodies. Nuclei were stained with Hoechst 33342 (Sigma-Aldrich, St. Louis, MO, USA). Mean fluorescence intensity (MFI) for PGC-1α and LARS1 of immunofluorescence images was quantified using Image J software (NIH; v.1.53t). To obtain region of interest (ROI) values of each marker in three different fields, the ‘Free Hand Selection’ mode was used. In addition, background ROI values were obtained. Then, the background ROI value was subtracted from three measured ROI values. The resulting values were averaged, and statistical analysis was performed.

### 2.13. LARS1 Expression and LARS1 shRNA Expression Vector Transfection

PGC-1α-HEK293, PGC-1α-1-SW480, PGC-1α shRNA-1-SW620, or NC shRNA-SW620 cells were transfected with NC shRNA (pGFP-V-RS shRNA vector; TR30007) or LARS1 shRNA expression vector (TG303577; Origene, Rockville, MD, USA) using lipofectamine 2000 according to the manufacturer’s instructions. In addition, PGC-1α shRNA-1-SW620 cells were transfected with 4 μg of LARS1 expression vector (LARS1 human tagged ORF clone; RG221682) or empty vector (pCMV6-AC-GFP; PS100010) using lipofectamine 2000 according to the manufacturer’s instructions. After transfection, cells were cultured in 10% FBS-supplemented DMEM for 48 h. These cells, pcDNA-HEK293, pcDNA-SW480, and NC shRNA SW620 cells were then used for cell counting followed by transwell migration and invasion assays.

### 2.14. PGC-1α Expression and PGC-1α shRNA Expression Vector Transfection

HT-29 cells were transfected with 4 μg of empty vector (pcDNA3.1) or pcDNA3.1-FLAG-PGC-1α expression vector from Spiegelman using Lipofectamine 2000 (Invitrogen, Carlsbad, CA, USA) following the manufacturer’s recommended procedure. After transfection, cells were cultured in 10% FBS-supplemented DMEM for 48 h. In addition, SNU-C4 cells were transfected with NC shRNA (pGFP-V-RS shRNA vector; TR30007) or PGC-1α shRNA expression vector (KH00461N; Qiagen, Hilden, Germany) using Lipofectamine 2000 (Invitrogen, Carlsbad, CA, USA) following the manufacturer’s instructions. These cells were then used for qRT-PCR and Western blot analysis.

### 2.15. Statistical Analysis

All statistical analyses were performed with PASWStatistics 18 software (SPSS, Chicago, IL, USA). Data are expressed as mean ± standard deviation (SD). One-way analysis of variance (ANOVA) and unpaired Student’s t-test were used to determine statistical significance. Statistical significance was defined as *p* < 0.05.

## 3. Results

### 3.1. Overexpression of PGC-1α Leads to Upregulation of LARS1 in Human Embryonic Kidney 293 (HEK293) Cells

Previously, we have reported that PGC-1α overexpression promotes cell proliferation and tumorigenesis of HEK293 cells [8]. In order to identify genes that increased proliferation and invasion by PGC-1α overexpression in HEK293 cells, ACP-based GeneFishing PCR was performed. Band densities between PGC-1α-overexpressed HEK293 cells (PGC-1α-HEK293 cells) and empty vector-expressing HEK293 cells (control HEK293 cells) were then compared. One fragment showing >2-fold different densities between two cell lines was observed by densitometric analysis of amplified cDNA fragments (Figure 1A). The amplified band was eluted from agarose gel, re-amplified, and sequenced. Sequences were used in a BLAST search to identify its gene annotations. BLAST analysis identified LARS1. In order to confirm increased expression of LARS1 in PGC-1α-HEK293 cells, qRT-PCR, Western blot analysis, and immunofluorescence staining were performed. The expression levels of LARS1 mRNA and protein were increased in PGC-1α-HEK293 cells (Figure 1B,C). These data suggest that PGC-1α may regulate the expression of LARS1.

### 3.2. LARS1 Overexpression Enhances Cell Proliferation, Migration, and Invasion of SW480 Cells

Previously, we observed that the expression of PGC-1α in human colorectal cancer SW620 cells was higher than in SW480 cells [10]. We established stable PGC-1α-overexpressing SW480 cells (PGC-1α-1- and PGC-1α-2-SW480 cells) and examined the mRNA levels of PGC-1α and LARS1 in these cells by qRT-PCR. As shown in Figure 2A, the expression levels of PGC-1α and LARS1 were significantly increased in PGC-1α-1- and -2 –SW480 cells. As we observed increased expression of LARS1 in PGC-1α-HEK293 cells, we examined expression levels of LARS1 in SW480 and SW620 cells. As expected, the expression level of LARS1 in SW620 cells was higher than in SW480 cells (Figure 2A). To investigate whether LARS1 expression affected cell proliferation, migration, and invasion of SW480 cells, we established stable LARS1-overexpressing SW480 cells (LARS1 low expression) and confirmed that LARS1 expression was increased in stable LARS1-overexpressing SW480 cells (LARS1-2-, -4-SW480 cells) by Western blot analysis (Figure 2A). We then examined cell proliferation by performing cell counting and MTT assay. As shown in Figure 2A,C, LARS1 overexpression enhanced cell proliferation in a time-dependent manner. In addition, results of transwell migration and invasion assays showed that numbers of migrating and invading LARS1-SW480 cells were significantly higher than those of control SW480 cells (Figure 2D,E).

### 3.3. LARS1 Knockdown Reduces Cell Proliferation, Migration, and Invasion of SW620 Cells

To confirm whether the LARS1 expression is regulated by PGC-1α expression in SW620 cells, we examined the mRNA levels of PGC-1α and LARS1 in PGC-1α shRNA-1-, -2-SW620 cells by qRT-PCR. As shown in Figure 3A, the expression levels of PGC-1α and LARS1 mRNAs were decreased in PGC-1α shRNA-1- and -2-SW620 cells. Since we observed that SW620 cells highly expressed LARS1 (Figure 2A), we established stable LARS1 shRNA-knocked down SW620 cells (LARS1 shRNA-4, -5-SW620 cells) and then confirmed that LARS1 expression was reduced in these stable LARS1 shRNA-knocked down SW620 cells by Western blot analysis (Figure 3A). However, PGC-1α expression levels in LARS1 shRNA-4-SW620 and LARS1 shRNA-5-SW620 cells were not changed compared to those in NC shRNA-SW620 cells, suggesting that PGC-1α acted upstream of LARS1. We then examined cell proliferation by performing cell counting and MTT assay. As shown in Figure 3B,C, LARS1 knockdown decreased cell proliferation in a time-dependent manner. In addition, results of transwell migration and invasion assays showed that numbers of migrating and invading cells of LARS1 shRNA-SW620 cells were significantly lower than those of NC shRNA-SW620 cells (Figure 3D,E).

### 3.4. PGC-1α Regulates Cell Proliferation, Migration, and Invasion of PGC-1α-HEK293, SW480, and SW620 Cells by Regulating LARS1 Expression

Based on the above observations, we hypothesized that PGC-1α could regulate cell proliferation, migration, and invasion through the modulation of LARS1 expression. To test this hypothesis, we examined the cell proliferation, migration, and invasion of PGC-1α-HEK293 and PGC-1α-1-SW480 cells by transfecting NC shRNA or LARS1 shRNA, respectively. As shown in Appendix A and Figure 4A, the cell proliferation of PGC-1α-HEK293 and PGC-1α-1-SW480 cells by NC shRNA transfection was higher than that of pcDNA-HEK293 and control SW480 (pcDNA-SW480) cells, respectively. The increase in cell proliferation of PGC-1α-HEK293 and PGC-1α-1-SW480 cells was decreased by LARS1 shRNA transfection, respectively. In addition, cell numbers of migrating and invading cells of PGC-1α-HEK293 and PGC-1α-1-SW480 cells after NC shRNA transfection were higher than those of pcDNA-HEK293 and control SW480 (pcDNA-SW480) cells, respectively. However, such increases of migrating and invading cells of PGC-1α-HEK293 and PGC-1α-1-SW480 cells were decreased by LARS1 shRNA transfection, respectively (Appendix A and Figure 4B,C). As shown in Figure 4D, cell proliferation of PGC-1α shRNA-1-SW620 cells after pCMV6 transfection was decreased compared to that of control SW620 (NC shRNA-SW620) cells. However, such a decrease in cell proliferation of PGC-1α-shRNA-1-SW620 cells was reversed by LARS1 transfection. In addition, numbers of migrating and invading cells of PGC-1α-shRNA-1-SW620 cells after pCMV6 transfection were decreased compared to those of control SW620 (NC shRNA-SW620) cells. However, such decreases in migrating and invading cells of PGC-1α-shRNA-1-SW620 cells were increased by LARS1 transfection (Figure 4E,F). As shown in Figure 4G, cell proliferation of PGC-1α-shRNA-1-SW620 cells after NC shRNA transfection was decreased compared to that of control SW620 (NC shRNA-SW620) cells. However, such a decrease in cell proliferation of PGC-1α shRNA-1-SW620 cells was not changed by LARS1 shRNA transfection. In addition, numbers of migrating and invading cells of PGC-1α shRNA-1-SW620 cells after NC shRNA transfection were decreased compared to those of control SW620 (NC shRNA-SW620) cells. However, such decreases of migrating and invading cells of PGC-1α shRNA-1-SW620 cells were not changed by LARS1 shRNA transfection (Figure 4H,I). These data indicate that effects of PGC-1α on cell proliferation and invasion are mediated by LARS1 expression.

### 3.5. PGC-1α Regulates Cell Proliferation, Migration, and Invasion through LARS1/AKT/GSK-3β/β-Catenin Pathway

Our previous study has demonstrated that PGC-1α can regulate cell proliferation and invasion through the AKT/GSK3β/β-catenin pathway [10]. The above observations provided some evidence suggesting that PGC-1α can regulate cell proliferation and invasion via the modulation of LARS1 expression. It has been shown that LARS1 regulates the mTORC1 pathway by activating Rag GTPase [12,13]. AKT is an upstream molecule of the mTORC1 pathway [27,28]. mTORC1 is known to phosphorylate its targets including S6K1 and 4EBP1 [16]. Thus, we examined the phosphorylation of AKT at the Ser473 site (active AKT), GSK-3β at the Ser9 site (inactive GSK-3β), mTOR at the Ser2448 site, S6K1 at the Thr389 site, 4EBP1 at the Thr37/46 site in pcDNA-SW480, PGC-1α-SW480 (PGC-1α-1, -2 -SW480), pCMV6-SW480, and LARS1-SW480 (LARS1-2, -4 -SW480) cells by Western blot analysis. As shown in Figure 5A,B, expression levels of p-AKT, p-GSK-3β, p-mTOR, p-S6K1, and p-4EBP1 were increased in PGC-1α-1, -2 -SW480, and LARS-2, -4 -SW480 cells than in pcDNA-SW480 and pCMV6-SW480 cells. In addition, expression levels of p-AKT, p-GSK-3β, p-mTOR, p-S6K1, and p-4EBP1 were reduced in PGC-1α shRNA-1, -2 -SW620 and LARS1 shRNA-4, -5 -SW620 cells than in NC shRNA-SW620 cells (Figure 5C,D). We also observed that expression levels of β-catenin, c-Myc, cyclin D1, and vimentin were increased in PGC-1α-1, -2 -SW480 and LARS1-2, -4 -SW480 cells than in pcDNA-SW480 and pCMV6-SW480 cells (Figure 5A,B). Expression levels of β-catenin, c-Myc, cyclin D1, and vimentin were decreased in PGC-1α shRNA-1, -2 -SW620 and LARS1 shRNA-4, -5 -SW620 cells than in NC shRNA-SW620 cells (Figure 5C,D). To confirm whether PGC-1α regulates the expression levels of LARS1, p-AKT, p-GSK-3β, p-mTOR, p-S6K1, p-4EBP1, β-catenin, c-Myc, cyclin D1, and vimentin in other colorectal cancer HT-29 and SNU-C4 cells, we performed qRT-PCR and Western blot analysis. The expression of PGC-1α and LARS1 in HT-29 cells were increased by PGC-1α transfection and were accompanied by the increased expression levels of p-AKT, p-GSK-3β, p-mTOR, p-S6K1, p-4EBP1, β-catenin, c-Myc, cyclin D1, and vimentin (Appendix A). In addition, the expression of PGC-1α and LARS1 in SNU-C4 cells were decreased by PGC-1α shRNA transfection and were accompanied by the decreased expression levels of p-AKT, p-GSK-3β, p-mTOR, p-S6K1, p-4EBP1, β-catenin, c-Myc, cyclin D1, and vimentin (Appendix A). These data support that PGC-1α regulates the LARS1/AKT/GSK-3β/β-catenin pathway.

As we observed activation of AKT in both PGC-1α-SW480 and LARS1-SW480 cells, we hypothesized that AKT activation was required for increased cell proliferation and invasion by LARS1 overexpression. To test this hypothesis, LARS1-2-SW480 cells were treated with AKT inhibitor IV. Cell proliferation. migration, and invasion were then examined. As shown in Figure 6A–D, increases of cell proliferation, migration, and invasion induced by LARS1 overexpression were reversed by AKT inhibitor IV. Increased expression levels of p-AKT, p-GSK-3β, β-catenin, c-Myc, cyclin D1, and vimentin by LARS1 overexpression were decreased by AKT inhibitor IV treatment (Figure 6E). Increased expression levels of p-mTOR, p-S6K1, and p-4EBP1 in LARS1-SW480 cells were also decreased by AKT inhibitor IV treatment (Figure 6E). However, expression levels of PGC-1α and LARS1 were not changed by AKT inhibitor IV treatment (Figure 6E). Additionally, we observed that increased cell proliferation and invasion in PGC-1α-HEK293 cells were significantly inhibited by AKT inhibitor IV (Appendix A). These data suggest that increases of proliferation, migration, and invasion induced by LARS1 overexpression are mediated by AKT activation and that PGC-1α acts upstream of LARS1 for regulating cell proliferation, migration, and invasion.

## 4. Discussion

There are still controversies about the role of PGC-1α in cancer, although intensive studies have been performed. Its actions and the underlying molecular mechanisms are very complicated depending on the cell type, its posttranslational modifications (such as phosphorylation, acetylation, ubiquitination, and so on), and the presence of its interacting proteins [4]. Our previous study has revealed that PGC-1α functions as a tumor promoter through the activation of AKT/GSK-3β/β-catenin signaling [10]. However, we did not clarify the underlying mechanism by which PGC-1α regulated the AKT/GSK-3β/β-catenin pathway, although we observed a direct interaction of PGC-1α and AKT by co-immunoprecipitation. In the present study, we identified LARS1 as one of candidate downstream targets of PGC-1α by using GeneFishing^TM^ DEG (Seegene, Seoul, Republic of Korea). LARS1 mRNA and protein expressions were increased in PGC-1α-HEK293, PGC-1α-1-, and -2-SW480 cells and in HT-29 cells by PGC-1α transfection, whereas the expressions of LARS1 mRNA and protein were decreased in PGC-1α shRNA-1- and -2-SW620 cells and in SNU-C4 cells by PGC-1α shRNA transfection. These results prompted us to hypothesize that the effects of PGC-1α on cell proliferation, migration, and invasion might be mediated by LARS1 expression. In order to investigate this possibility, first we established stable LARS1-overexpressing SW480 cells (LARS1-SW480; LARS1-2- and -4-SW480) and stable LARS1 shRNA-knocked down SW620 (LARS1 shRNA-SW620; LARS1 shRNA-4- and -5-SW620) cells and found that LARS1-SW480 cells showed higher cell proliferation, migration, and invasion than pCMV6-SW480 cells. In addition, LARS1 shRNA-SW620 cells showed less cell proliferation, migration, and invasion than NC shRNA-SW620 cells. These findings are consistent with previous reports showing that LARS1 expression is related to the growth and migration of lung cancer cells, is essential for survival in osteosarcoma, and is required for cell proliferation in tuberous sclerosis complex (TSC)–null cells [17,29,30]. However, we did not examine the expression of TSC in SW480 or SW620 cells. Further studies are needed to investigate the expression of TSC1/2 in SW480 and SW620 cells. Very surprisingly, in contrast to our results, Passarelli et al. have demonstrated that LARS1 has a tumor suppressive effect by repressing the codon-dependent translation of epithelial membrane protein 3 (EMP3) and gamma-glutamyltransferase 5 (GGT5) in breast cancer [31]. These results implicate that the effects of LARS1 on tumor progression are variable depending on the cellular context.

As previously mentioned, the effects of LARS1 on cell proliferation and invasion are very similar to those of PGC-1α. Our mechanistic studies showed that LARS1 was required for effects of PGC-1α on cell proliferation and invasion using LARS1 shRNA knockdown or LARS1 overexpression in PGC-1α-SW480 cells and PGC-1α shRNA-SW620 cells, respectively. In addition, decreased cell proliferation and invasion in PGC-1α shRNA-SW620 cells were not changed by LARS1 shRNA transfection. These results support the hypothesis that PGC-1α regulates cell proliferation and invasion via LARS1. Interestingly, we have found that the LARS1 promoter/enhancer has a Sp1 binding site using GeneCards (http://genecards.org; accessed on 22 December 2022), and we have previously shown that PGC-1α enhances cell proliferation and tumorigenesis of HEK293 cells through the upregulation of Sp1 and ACBP [8]. This suggests that LARS1 expression is likely to be regulated by PGC-1α via Sp1. In the near future, we will investigate whether PGC-1α increases the expression of LARS1 via the binding of Sp1 to the promoter of LARS1. Activation of the AKT/GSK-3β/β-catenin axis, mTOR, S6K1, and 4EBP1 was observed in both PGC-1α-SW480 and LARS1-SW480 cells, whereas inhibition of those molecules was observed in both PGC-1α shRNA- and LARS1 shRNA-SW620 cells.

GSK-3β is a negative regulator of β-catenin [32]. Increasing evidence has shown that GSK-3β is a downstream molecule of AKT. It can be inactivated by AKT-mediated phosphorylation at its Ser9 residue [33,34,35]. Previous studies and the present results demonstrated that PGC-1α/LARS1 overexpression upregulated whereas PGC-1α/LARS1 knockdown downregulated phosphorylated (activated) AKT and phosphorylated (inactivated) GSK-3β levels. Furthermore, inhibition of AKT by AKT inhibitor IV in LARS1-SW480 cells reversed the ability of LARS1 to induce GSK-3β inactivation, β-catenin pathway activation, cell proliferation, migration, and invasion. These data reveal that LARS1-mediated AKT activation can lead to inactivation of GSK-3β and consequent β-catenin pathway activation. The present study suggests that AKT activation is required for increased cell proliferation and invasion by LARS1 overexpression. However, our results showed that LARS1 activated ATK, different from other studies demonstrating that LARS1 expression is inversely related to p-AKT in non-small cell lung cancer cell lines and tissues [19] and that LARS1 inhibits myogenic differentiation through the Rag-mTORC1 pathway and subsequent inhibition of the IRS1-PI3K-AKT pathway [36]. These differential effects of LARS1 on the activation of AKT are dependent on the cellular context.

Our previous study has demonstrated that decreased cell proliferation and invasion by PGC-1α knockdown are reversed by constitutive AKT expression [10]. Additionally, our present findings suggest that PGC-1α acts upstream of LARS1. Altogether, these results implicate that PGC-1α regulates cell proliferation and invasion through modulation of the LARS1/AKT/GSK3β/β-catenin axis and that LARS1 might be a potential therapeutic target for PGC-1α-overexpressing human colorectal cancer as depicted in a schematic diagram (Figure 7). However, our study has some limitations. First, we did not evaluate the in vivo effect of the PGC-1α/LARS1 axis on tumor progression of human colorectal cancer. Thus, further research is warranted in the context of using in vivo models to evaluate whether modulation of LARS1 expression by LARS1 overexpression or LARS1 knockdown can reverse PGC-1α actions on tumor progression. Second, the underlying molecular mechanisms by which PGC-1α regulates LARS1 and how LARS1 regulates AKT were not clarified in this study. Thus, further exploration is needed to dissect its detailed mechanisms. Third, we did not evaluate the crosstalk between mTORC1 and GSK-3β in this study. Further studies about the molecular network between mTORC1 and GSK-3β are required in the future.

## 5. Conclusions

To the best of our knowledge, this is the first report showing that PGC-1α regulates cell proliferation and invasion through modulation of LARS1 expression. Further studies evaluating the clinical significance of the PGC-1α/LARS1/AKT/GSK-3β/β-catenin axis in human colorectal cancer patients and therapeutic potentials of several LARS1 inhibitors in PGC-1α-overexpressing colorectal cancer are needed.

## Figures and Tables

**Figure 1 cancers-15-00159-f001:**
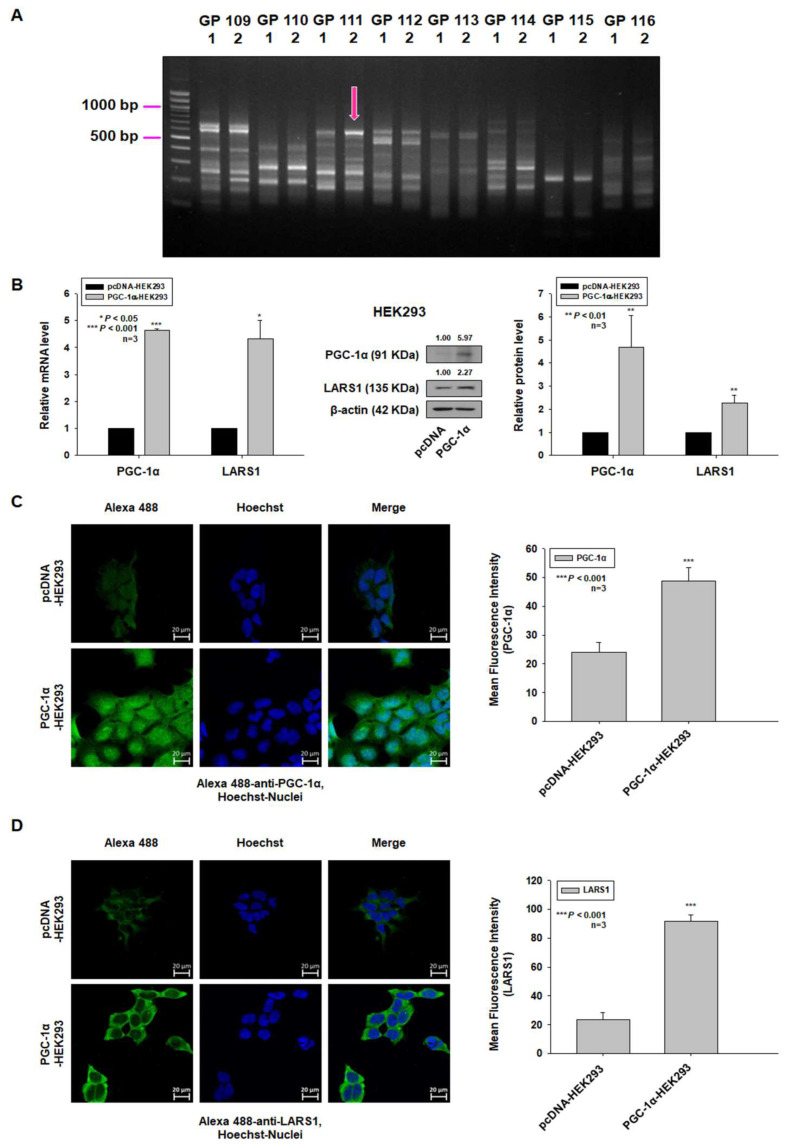
PGC-1α overexpression enhances expression of LARS1 in HEK293 cells. (**A**) Total RNAs isolated from PGC-1α- or pcDNA-expressing HEK293 cells were used for ACP-based GeneFishing analysis as described in Materials and Methods. Amplified bands showing different densities between two cell lines were re-amplified and sequenced for gene annotation. Arrow indicates DEG band identified as LARS1. (**B**) Left: The mRNA levels of PGC-1α and LARS1 were examined by qRT-PCR. β-actin is used as an endogenous control. Middle and Right: The protein levels of PGC-1α and LARS1 in control pcDNA-HEK293 and PGC-1α-HEK293 cells. Protein lysates were prepared and subjected to Western blot analysis as described in Materials and Methods. Equal protein loading was ensured by showing uniform β-actin expression. Densitometry results are indicated above the bands. Representative data of three independent experiments are shown. Data are expressed as the mean ± SD of three independent experiments. * *p* < 0.05, ** *p* < 0.01, *** *p* < 0.001, pcDNA-HEK293 cells. (**C**,**D**) Left panel: Immunofluorescence staining was performed as described in Materials and Methods using anti-PGC-1α ((**C**), green) and anti-LARS1 ((**D**), green) antibodies. Right panel: Mean Fluorescence Intensity (MFI) of PGC-1α (**C**) and LARS1 (**D**) of immunofluorescence images were quantified using Image J software (v.1.53t). GP, arbitrary general primer; 1, control HEK293 cells; 2, PGC-1α-HEK293 cells. Molecular weights for proteins are indicated in the full, uncropped, annotated Western blot images (Appendix A).

**Figure 2 cancers-15-00159-f002:**
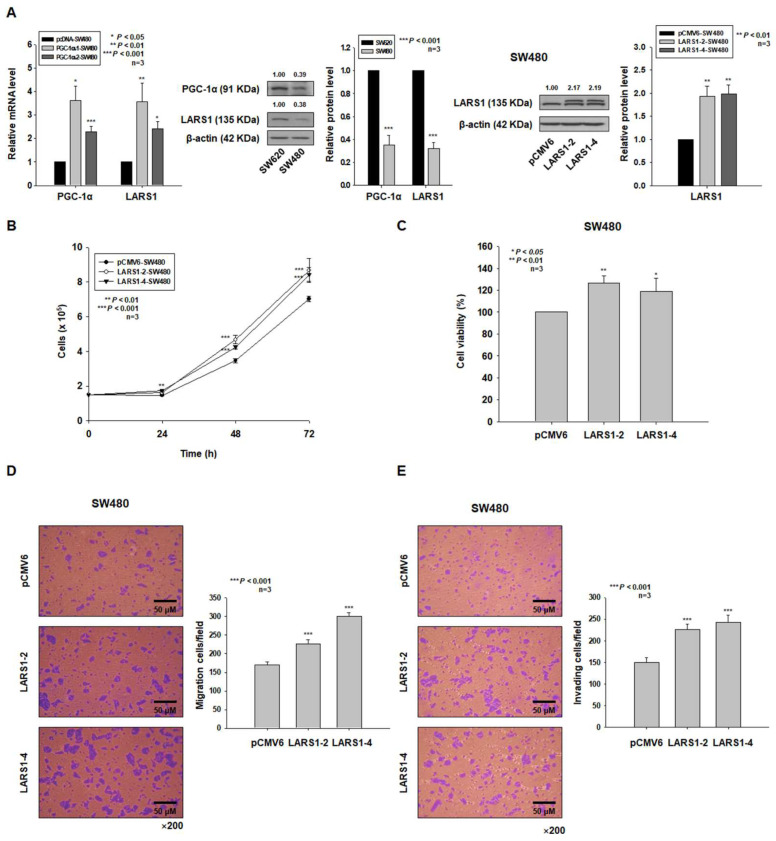
LARS1 overexpression enhances cell proliferation, migration, and invasion of SW480 cells. (**A**) Left panel: Expression levels of PGC-1α and LARS1 mRNA in pcDNA-, PGC-1α-1-, or -2-SW480 cells. Middle panel: Expression of PGC-1α and LARS1 in SW620 and SW480 cells. Right panel: SW480 cells were transfected with LARS1 or pCMV6 expression vector and screened based on their resistance to G418 (800 μg/mL). Western blot was used to detect PGC-1α and LARS1. β-actin was used as an internal control. Densitometry results are indicated above bands. Representative data of three independent experiments are shown. Data are expressed as the mean ± SD of three independent experiments. *** *p* < 0.001 vs. pcDNA-SW480 cells. (**B**,**C**) pCMV6-, LARS1-2, or -4-SW480 cells were seeded and cultured for the indicated times and cell proliferation was determined by cell counting (**B**) and MTT assay (**C**). Data are presented as the mean ± SD of three separate experiments. * *p* < 0.05, ** *p* < 0.01, *** *p* < 0.001 vs. pCMV6-SW480 cells. (**D**,**E**) Left panel: Representative figures of pCMV6-, LARS1-2-, and -4-SW480 cells in the transwell migration assay (**D**) and transwell invasion assay (**E**) are shown (×200 magnification). Right panel: The numbers of transmembrane migrated cells (**D**) and transmembrane invaded cells (**E**) were counted for five randomly chosen visual fields. Data are expressed as the mean ± SD of three independent experiments. *** *p* < 0.001 vs. pCMV6-SW480 cells. Molecular weights for proteins are indicated in the full, uncropped, annotated Western blot images (Appendix A).

**Figure 3 cancers-15-00159-f003:**
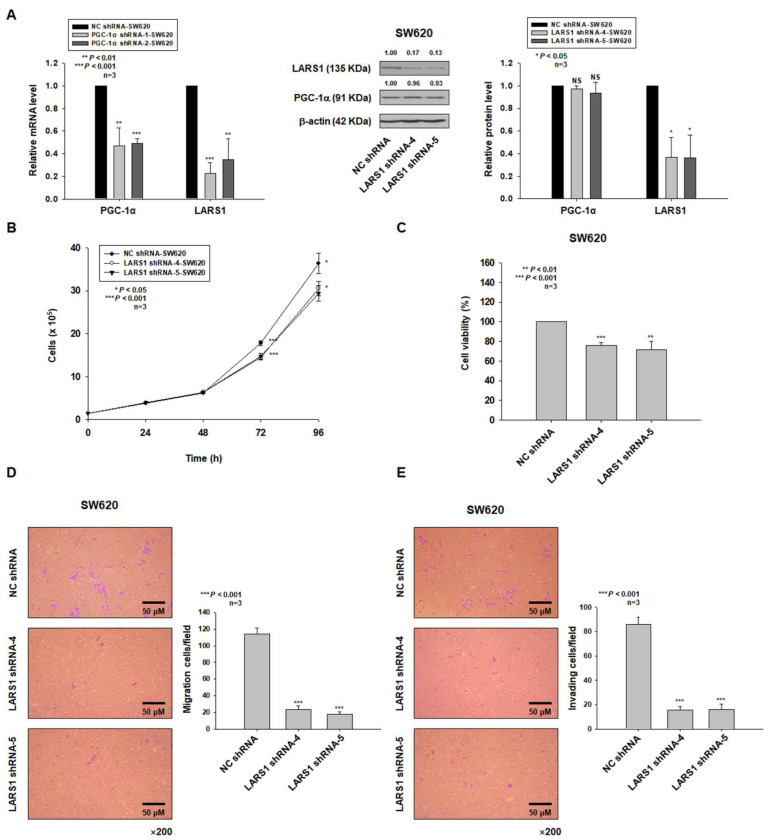
LARS1 knockdown reduces cell proliferation, migration, and invasion of SW620 cells. (**A**) Left panel: Expression levels of PGC-1α and LARS1 mRNA in NC shRNA-, PGC-1α shRNA-1-, or -2-SW620 cells. Data are expressed as the mean ± SD of three independent experiments. ** *p* < 0.01, *** *p* < 0.001, NC shRNA-SW620 cells. Middle panel: SW620 cells were transfected with NC shRNA or LARS1 shRNA expression vector and screened based on their resistance to G418 (800 μg/mL). Western blot was used to detect PGC-1α and LARS1. β-actin was used as an internal control. Densitometry results are indicated above bands. Representative data of three independent experiments are shown. Right panel: Data are expressed as the mean ± SD of three independent experiments. * *p* < 0.05 vs. NC shRNA-SW620 cells. NS, not significant. (**B**,**C**) NC shRNA-, LARS1 shRNA-4-, or -5-SW620 cells were seeded and cultured for the indicated time and cell proliferation was determined by cell counting (**B**) and MTT assay (**C**). Data are presented as the mean ± SD of three separate experiments. * *p* < 0.05, ** *p* < 0.01, *** *p* < 0.001 vs. NC shRNA-SW620 cells. (**D**,**E**) Left panel: Representative figures of NC shRNA- and LARS1 shRNA-5-SW620 cells in the transwell migration assay (**D**) and transwell invasion assay (**E**) are shown (×200 magnification). Right panel: The numbers of transmembrane migrated cells (**D**) and transmembrane invaded cells (**E**) are shown. Data are expressed as the mean ± SD of three independent experiments. *** *p* < 0.001 vs. NC shRNA-SW620 cells. Molecular weights for proteins are indicated in the full, uncropped, annotated Western blot images (Appendix A).

**Figure 4 cancers-15-00159-f004:**
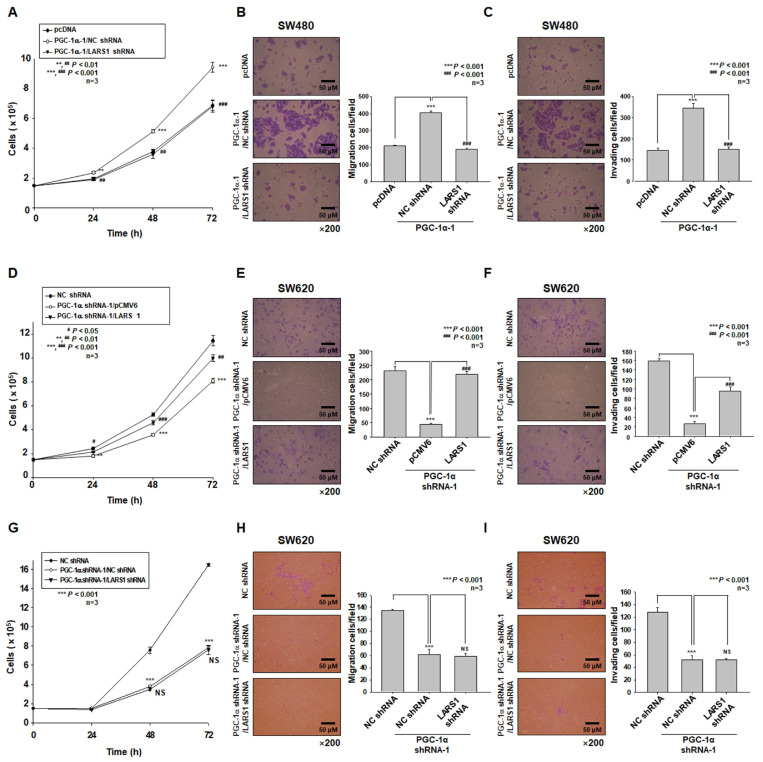
PGC-1α regulates cell proliferation, migration, and invasion of SW480 and SW620 cells by regulating LARS1 expression. (**A**–**C**) PGC-1α-1-SW480 cells were transiently transfected with NC shRNA vector or LARS1 shRNA expression vector, respectively. (**A**) After transfection, the cell counts of each type of cell are presented as the mean ± SD of three separate experiments. (**B**,**C**) Left panel: Representative figures of pcDNA-, PGC-1α-1/NC shRNA-, and PGC-1α-1/LARS1 shRNA-SW480 cells in the transwell migration assay (**B**) and transwell invasion assay (**C**) are shown (×200 magnification). Right panel: The numbers of transmembrane migrated cells (**B**) and transmembrane invaded cells (**C**) are shown. Data are expressed as the mean ± SD of three independent experiments. ** *p* < 0.01, *** *p* < 0.001 vs. pcDNA-SW480 cells. ^##^ *p* < 0.01, ^###^ *p* < 0.001 vs. PGC-1α-1/NC shRNA-SW480 cells. (**D**–**F**) PGC-1α shRNA-1-SW620 cells were transiently transfected with pCMV6 expression vector or LARS1 expression vector, respectively. (**D**) After transfection, the cell counts of each type of cell are presented as the mean ± SD of three separate experiments. *** *p* < 0.001 vs. NC shRNA-SW620 cells. ^#^ *p* < 0.05, ^##^ *p* < 0.01, ^###^ *p* < 0.001 vs. PGC-1α shRNA-1/pCMV6-SW620 cells. (**E**,**F**) Left panel: Representative figures of NC shRNA-, PGC-1α shRNA-1/pCMV6-, and PGC-1α shRNA-1/LARS1-SW620 cells in the transwell migration assay (**E**) and transwell invasion assay (**F**) are shown (×200 magnification). Right panel: The numbers of transmembrane migrated cells (**E**) and transmembrane invaded cells (**F**) are shown. Data are expressed as the mean ± SD of three independent experiments. *** *p* < 0.001 vs. NC shRNA-SW620 cells. ^#^ *p* < 0.05, ^##^ *p* < 0.01, ^###^ *p* < 0.001 vs. PGC-1α shRNA-1/pCMV6-SW620 cells. (**G**–**I**) PGC-1α shRNA-1-SW620 cells were transiently transfected with NC shRNA expression vector or LARS1 shRNA expression vector, respectively. (**G**) After transfection, the cell counts of each type of cell are presented as the mean ± SD of three separate experiments. (**H**,**I**) Left panel: Representative figures of NC shRNA-, PGC-1α shRNA-1/NC shRNA-, and PGC-1α shRNA-1/LARS1 shRNA-SW620 cells in the transwell migration assay (**H**) and the transwell invasion assays (**I**) are shown (×200 magnification). Right panel: The numbers of transmembrane migrated cells (**H**) and transwell-membrane invaded cells (**I**) are shown. Data are expressed as the mean ± SD of three independent experiments. *** *p* < 0.001 vs. NC shRNA-SW620 cells. NS, not significant.

**Figure 5 cancers-15-00159-f005:**
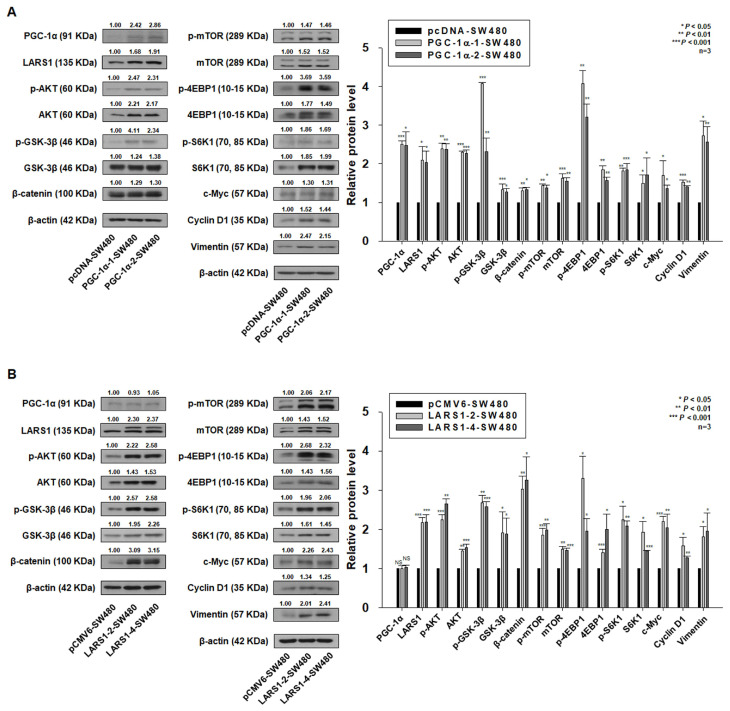
Expression levels of PGC-1α, LARS1, and several signaling molecules in pcDNA-, PGC-1α-, pCMV6-, LARS1-SW480, NC shRNA-, PGC-1α shRNA-, LARS1 shRNA-SW620 cells. (**A**) (Left panel) Representative three independent Western blot analysis results of pcDNA-, PGC-1α-1-, PGC-1α-2-SW480 cells are shown. β-actin was used as a loading control. Densitometry results are expressed above bands. (Right panel) Data are expressed as the mean ± SD of three independent experiments. * *p* < 0.05, ** *p* < 0.01, *** *p* < 0.001 vs. pcDNA-SW480 cells. (**B**) (Left panel) Representative three independent Western blot analysis results of pCMV6-, LARS1-2-, LARS1-4-SW480 cells are shown. β-actin was used as a loading control. Densitometry results are expressed above the bands. (Right panel) Data are expressed as the mean ± SD of three independent experiments. * *p* < 0.05, ** *p* < 0.01, *** *p* < 0.001 vs. pCMV6-SW480 cells. (**C**) (Left panel) Representative three independent Western blot analysis results of NC shRNA-, PGC-1α shRNA-1-, PGC-1α shRNA-2-SW620 cells are shown. **β**-actin was used as a loading control. Densitometry results are expressed above bands. (Right panel) Data are expressed as the mean ± SD of three independent experiments. * *p* < 0.05, ** *p* < 0.01, *** *p* < 0.001 vs. NC shRNA-SW620 cells. (**D**) (Left panel) Representative three independent Western blot analysis results of NC shRNA-, LARS1 shRNA-4-, LARS1 shRNA-5-SW620 cells are shown. β-actin was used as a loading control. Densitometry results are expressed above bands. (Right panel) Data are expressed as the mean ± SD of three independent experiments. * *p* < 0.05, ** *p* < 0.01, *** *p* < 0.001 vs. NC shRNA-SW620 cells. Molecular weights for proteins are indicated in the full, uncropped, annotated Western blot images (Appendix A).

**Figure 6 cancers-15-00159-f006:**
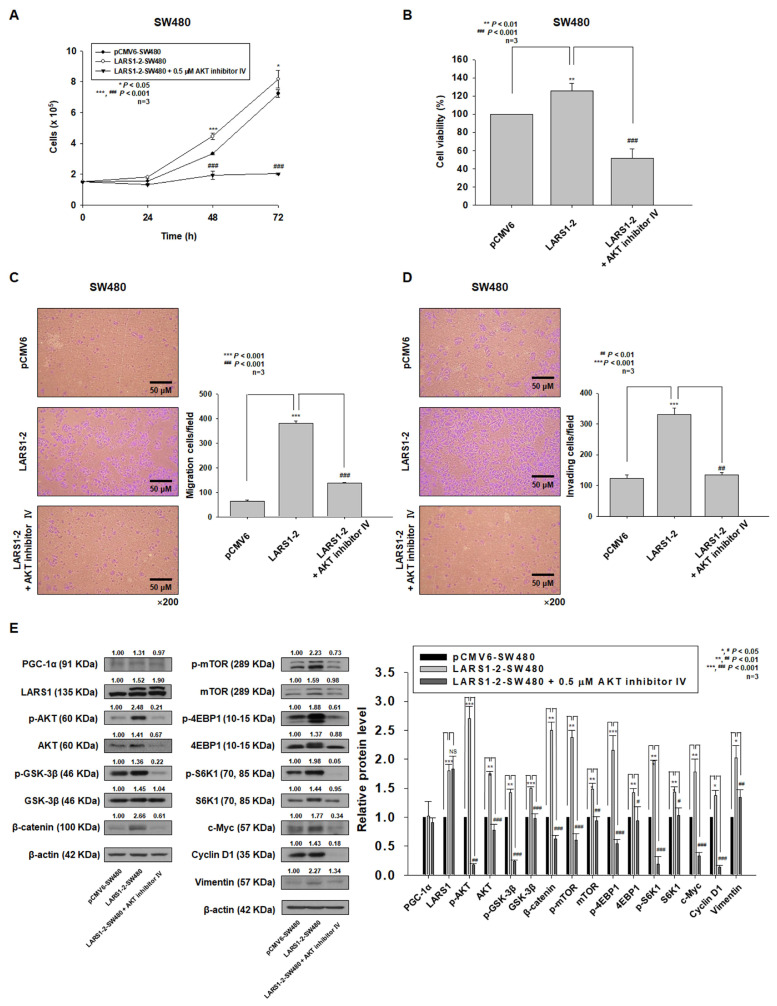
LARS1 enhances cell proliferation, migration, and invasion through AKT activation in SW480 cells. (**A**,**B**) LARS1-2-SW480 cells were treated with/without 0.5 μM AKT inhibitor IV for indicated time (**A**) or for 48 h (**B**) and cell proliferation was measured by cell counting (**A**) and MTT assay (**B**), respectively. (**C**,**D**) LARS1-2-SW480 cells were treated with/without 0.5 μM AKT inhibitor IV for 48 h. Transwell migration assays (**C**) and invasion assays (**D**) were performed as described in Materials and Methods. (**C**) Left panel: Representative figures of pCMV6-, LARS1-2-SW480 cells treated without/with 0.5 μM AKT inhibitor IV in the transwell migration assay are shown (×200 magnification). Right panel: The number of transmembrane migrated cells was counted for five randomly chosen visual fields. Data are expressed as the mean ± SD of three independent experiments. *** *p* < 0.001 vs. pCMV-6-SW480 cells; ^###^ *p* < 0.001 vs. LARS1-2-SW480 cells. (**D**) Left panel: Representative figures of pCMV6-, LARS1-2-SW480 cells treated without/with 0.5 μM AKT inhibitor IV in the transwell invasion assay are shown (×200 magnification). Right panel: The number of transmembrane migrated cells was counted for five randomly chosen visual fields. Data are expressed as the mean ± SD of three independent experiments. *** *p* < 0.001 vs. pCMV-6-SW480 cells; ^###^ *p* < 0.001 vs. LARS1-2-SW480 cells. (**E**) After treatment with AKT inhibitor IV, protein lysates were prepared and used for Western blot analysis with corresponding antibodies. β-actin was used as a loading control. (Left panel) The blot is representative of three separate experiments. Densitometry results are expressed above bands. (Right panel) Data are expressed as the mean ± SD of three independent experiments. * *p* < 0.05, ** *p* < 0.01, *** *p* < 0.001 vs. pCMV6-SW480 cells. ^#^ *p* < 0.05, ^##^ *p* < 0.01, ^###^ *p* < 0.001 vs. LARS1-2-SW480 cells. Molecular weights for proteins are indicated in the full, uncropped, annotated Western blot images (Appendix A).

**Figure 7 cancers-15-00159-f007:**
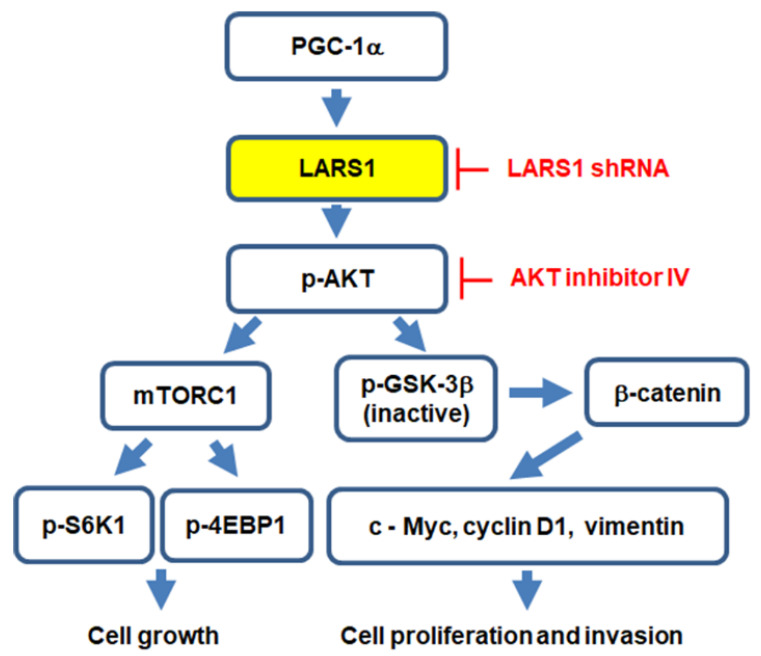
Potential molecular mechanism by which PGC-1α regulates cell proliferation, migration, and invasion of human colorectal cancer cells. In summary, PGC-1α regulates cell proliferation, migration, and invasion via regulation of LARS1/AKT/GSK-3β/β-catenin axis.

## Data Availability

All data generated or analyzed during this study are available from the corresponding author upon reasonable request.

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
