# Peer review of "PGC-1α Regulates Cell Proliferation, Migration, and Invasion by Modulating Leucyl-tRNA Synthetase 1 Expression in Human Colorectal Cancer Cells"

_cancers, 2022, doi:10.3390/cancers15010159_

Round 1

Reviewer 1 Report

Basic Reporting and comments

This study describes the role of PGC-1alpha regulates the cell proliferation through LARS/AKT axis in human colorectal cancer cells.

1. Typo in line 116

2. Transwell migration and invasion assay is one of the important assays related to manuscript. That should be explained in detail in the methodology section.

3.Overall in all the figures, picture quality is poor, with very small font sizes it is not clear to understand the phenomena going on. Contrast has to be improved.  All the pictures have to be revised for better readability.

4. Font sizes in all the figures have to be increased.

5. Figure legends can be made smaller if there is space constraint as they are too big. Extra information can be put into the supplementary material.

6. Scale bar is missing in all the microscopic images in all the figures.

7. Very poor representation is done in the bar graphs and western blots in figure 5. Nothing is visible.

8. Nothing is visible in the Figure 6C and D in the microscopic images. In nutshell, all images have to be taken with better contrast again. 

Author Response

We would like to thank the reviewers for their helpful and thoughtful comments. According to the reviewers’ comments, we have made changes in the manuscript, as described below. Changes are indicated with track change in the revised version of the manuscript.

  1. Typo in line 116

Response) We corrected the typo in line 116.

  1. Transwell migration and invasion assay is one of the important assays related to manuscript. That should be explained in detail in the methodology section.

Response) According to the reviewer’s comments, we have described the transwell and migration assays in detail in the revised manuscript.

    3. Overall in all the figures, picture quality is poor, with very small font sizes it is not clear to understand the phenomena going on. Contrast has to be improved. All the pictures have to be revised for better readability.

Response) We have improved the quality of the figures and increased the font size in all the figures as the reviewer suggested.

  1. Font sizes in all the figures have to be increased.

Response) According to the reviewer’s comments, we have increased the font sizes in all the figures.

  1. Figure legends can be made smaller if there is space constraint as they are too big. Extra information can be put into the supplementary material.

Response) We have shortened the figure legends acording to the reviewer’s suggestion.

  1. Scale bar is missing in all the microscopic images in all the figures.

Response) We have added the scale bar in all the microscopic images in all the figures, according to the reviewer’s comments.

  1. Very poor representation is done in the bar graphs and western blots in figure 5. Nothing is visible.

Response) We have improved the quality of bar graphs and western blots in figure 5 according to the reviewer’s suggestion.

  1. Nothing is visible in the Figure 6C and D in the microscopic images. In nutshell, all images have to be taken with better contrast again.

Response) According to the reviewer’s comments, we have improved the microscopic images and all images.

Reviewer 2 Report

Current manuscript explained the role of LARS in controlling Human Colorectal Cancer working in tandem with PGC-1alfa. Most of the part of the manuscript is well written and explained accordingly, but still some limitations are there and needs to be address.

1) There is no experimental evidence or even not properly explained how PGC-1alfa regulates LARS expression, as well as how LARS expression regulates AKT function.

2) Figure 1A needed explanations regarding header, red arrow needs to move at different angle to make clear visualization of next lane. It is also recommended to include q-PCR data for LARS expression under influence of PGC-1alfa for all of the tested cells.

3) All the western-blot of Figure 5 needed to enlarge in size as it is impossible to read at current form.

Author Response

1) There is no experimental evidence or even not properly explained how PGC-1alfa regulates LARS expression, as well as how LARS expression regulates AKT function.

Response) We thank the reviewer for pointing this out and we agree with the reviewer. Therefore, we have explained the potential mechanism how PGC-1a regulates LARS1 expression in the “Discussion” section. Our previous study has shown that PGC-1a enhances cell proliferation and tumorigenesis of HEK293 cells through the upregulation of Sp1 and ACBP (Shin et al., Int J Oncol. 2015). Thus, we have searched whether the promoter/enhancer of LARS1 has a Sp1 binding site using GeneCards (http://genecards.org). We have found that LARS1 promoter/enhancer has a Sp1 binding site. Thus, the upregulation of Sp1 by PGC-1a might be attributed to increased expression of LARS1. In the near future, we will investigate whether PGC-1a increases the expression of LARS1 via the binding of Sp1 to the promoter of LARS1. We did not investigate how LARS1 expression regulates AKT function in this study. Thus, further exploration is needed to dissect its detailed mechanisms. We have described it as a limitation of our study in the “Discussion” section.   

2) Figure 1A needed explanations regarding header, red arrow needs to move at different angle to make clear visualization of next lane. It is also recommended to include q-PCR data for LARS expression under influence of PGC-1alfa for all of the tested cells.

Response) As we have described in Methods section, Annealing control primer (ACP)-based GeneFishing PCR has been performed using 120 different types of arbitrary ACPs (arbitrary general primer; GP). We have described it in Figure 1 legend as a GP (arbitrary general primer). According to the revierwer’s comments, we have moved the red arrow at different angle. We have added the qRT-PCR data for LARS1 expression under influence of PGC-1a for all of the tested cells in Figure 1B, Figure 2A, Figure 3A, and Supplementary Figure S2.

3) All the western-blot of Figure 5 needed to enlarge in size as it is impossible to read at current form.

Response) According to the reviewer’s comments, we have enlarged the size of all the western blot of Figure 5.

Reviewer 3 Report

The manuscript, "PGC-1( Regulates Cell Proliferation, Migration, and Invasion by Modulating Leucyl-tRNA Synthetase Expression in Human Colorectal Cancer Cells, " was presented as an original research article where Cho et al. showed that PGC-1a regulates cell proliferation and invasion by controlling LARS/AKT/GSK3b/b-catenin axis in human colorectal cancer cells. The manuscript is well structured and supported by proper experiments, though I have a few minor comments that require attention and clarification, as detailed below.

1-     Did the authors check the possible synergistic effect of Knocking down both PGC1 and LARS? That can give an insight into whether PGC-1 regulates cell proliferation and invasion via LARS or has additional pathways.

2-     Including TOP/FOP flash assay could show the overall status (activation/inactivation) of canonical Wnt/Beta-catenin signaling in CRC cell lines.

3-     In Figure 1 B, the second panel, I can not see a significant difference in the expression of LARS. Less loading of gel samples could show a considerable difference.

4-     The quantification data must be included for IF images in figure1. The scale bars should be given contrasting colors so they can be seen clearly.

Author Response

1-Did the authors check the possible synergistic effect of Knocking down both PGC1 and LARS? That can give an insight into whether PGC-1 regulates cell proliferation and invasion via LARS or has additional pathways.

Response) We thank the reviewer for pointing this out and we agree with reviewer. We have examined the effect of LARS1 knockdown, by transfection using LARS1 shRNA, on cell proliferation and invasion in PGCa shRNA-SW620 cells. As results, LARS1 knockdown did not change the decreased cell proliferation and invasion in PGCa shRNA-SW620 cells. Thus, these results support the hypothesis that PGC-1a regulates cell proliferation and invasion via LARS1. These data are added to Figure 4G, 4H, amd Figure 4I. 

2-Including TOP/FOP flash assay could show the overall status (activation/inactivation) of canonical Wnt/Beta-catenin signaling in CRC cell lines.

Response) We tried to get TOP/FOP flash assay kit according to the reviewer’s comments. However, it takes at least more than 8 weeks to get it. Thus, we could not perform the experiments during the revision time. Please understand our situation. In addition, we have observed the expression levels of b-catenin, c-myc, vimentin, cyclin D1 in SW480 and SW620 cells, and their expresseion levels may reflect the canonical Wnt/b-catenin signaling in these cells.  

3-In Figure 1 B, the second panel, I can not see a significant difference in the expression of LARS. Less loading of gel samples could show a considerable difference.

Response) We have changed the data according to the reviewer’s suggestion, by less loading the gel samples.

4-The quantification data must be included for IF images in figure1. The scale bars should be given contrasting colors so they can be seen clearly.

Response) According to the reviewer’s comments, we have added the quantification data for IF images in figure 1 and changed the color of scale bars so they can be seen clearly.
